

# C/EBPα involvement in microglial polarization via HDAC1/STAT3 pathway aggravated sevoflurane-induced cognitive impairment in aged rats

Zhao Xu, Xi Yao, Yikang Zhao and Bo Yao

Department of Anesthesiology, Shaanxi Provincial People's Hospital, Xi'an, China

## ABSTRACT

**Background.** Postoperative cognitive dysfunction (POCD) is a clinically frequent postoperative complication in the elderly, which is mainly manifested by the occurrence of cognitive dysfunction after anesthetized surgery in patients. To explore the involvement of C/EBPα in microglial polarization in sevoflurane anesthesia induced cognitive impairment in aged rats.

**Methods.** Sprague-Dawley (SD) rats were anesthetized by inhalation of 3% sevoflurane for 6 h to establish the POCD model. The histopathological structure of hippocampus was observed by hematoxylin and eosin (HE) staining. Associative learning and memory function and spatial learning and memory function were assessed by conditioned fear test and water maze test. The concentrations of inflammatory factors in the hippocampus were measured by ELISA. The levels of microglial activation marker (Iba1) and microglial M1 (CD86) and M2 (CD206) polarization markers were determined by immunofluorescence staining and RT-qPCR, respectively. The transcriptional regulation of HDAC1 by C/EBPα was confirmed by dual luciferase reporter assay and ChIP assay.

**Results.** Sevoflurane-induced pathomorphological damage in the hippocampal tissue of aged rats, accompanied by elevated expression of C/EBPα. Silencing of C/EBPα alleviated hippocampal histopathological injury, inhibited M1 microglial activation and the expression of M1 marker CD86, enhanced the expression of M2 marker CD206. C/EBPα transcriptionally activated HDAC1. Knockdown of C/EBPα downregulated the expression of HDAC1 and STAT3 phosphorylated proteins, which inhibited the pro-inflammatory factors (IL-6 and TNF-α) and accelerated anti-inflammatory factors (IL-10 and TGF-β) secretion. In addition, silencing of C/EBPα caused rats to have a delayed freezing time in contextual conditioned fear, a shorter escape latency, and an increased number of platform crossings.

**Conclusion.** Inhibition of C/EBPα promotes the M2 polarization of microglia and reduces the production of pro-inflammatory cytokines to alleviate the cognitive dysfunction of sevoflurane-induced elderly rats by HDAC1/STAT3 pathway.

Corresponding author
Bo Yao, fredcarson@163.com

## INTRODUCTION

Postoperative cognitive dysfunction (POCD) is a clinical complication that often occurs in elderly patients after surgery. POCD mainly occurs 5–7 days after surgery, and patients older than 65 years have a short-term POCD incidence of approximately 20%–40% and a long-term POCD incidence of approximately 10% after undergoing noncardiac surgery (*Evered & Silbert, 2018*; *Berger et al., 2018*). Cognitive decline in older patients after anaesthetic surgery is characterized by mild thought impairment, inability to concentrate, memory impairment, and reduced ability to learn executive and language comprehension. Although the pathophysiological mechanism of POCD is still unclear. Studies increasingly believe that aging has become an important and independent risk factor for POCD (*Lin et al., 2020*). Age related brain decline can produce a variety of behavioral defects (*Mufson et al., 2016*; *Thomas, 2016*). Sevoflurane is a widely used inhalation anesthetic in clinical practice. Studies have shown that sevoflurane can activate neuroinflammation, aggravate nervous system damage, and lead to long-term neurocognitive dysfunction. (*Shao et al., 2020*; *Chen et al., 2020*). Therefore, it is of great practical importance to elucidate the mechanism of sevoflurane neurotoxicity and may help to prevent cognitive dysfunction.

Microglia, an important component in the central nervous system (CNS), belong to the type of glial cells, which function similarly to macrophages and are considered the most primary one of the immune defenses in the CNS. Accumulating evidence suggests that microglial activation in the CNS is heterogeneous and can be divided into two opposite types: M1 phenotype and M2 phenotype (*Tang & Le, 2016*; *Guo, Wang & Yin, 2022*). Activated microglia can play dual roles depending on the activated phenotype, therefore, microglia can produce either cytotoxic M1 or neuroprotective M2. When microglia achieve an M1 phenotype, large amounts of inflammatory cytokine products including IL-1β, TNF-α, IL-6 are produced. However, the anti-inflammatory cytokines IL-4, IL-10, IL-13, and TGF-β, among others, are released when the M2 phenotype is present. Morphological changes in activated microglia and an increase in inflammatory factors are observed in the aging brain. Morphological alterations have also been suggested as a potential mechanism for age-related neurodegeneration (*Crotti & Ransohoff, 2016*; *Costa et al., 2021*). It has been confirmed that neuroinflammation leads to the occurrence of POCD by promoting the release of inflammatory factors in the periphery and the disruption of the blood–brain barrier, reaching the center to cause the occurrence of central inflammation as well as the apoptosis of neurons. Previous studies have shown that sevoflurane promotes microglial M1 activation suppresses microglial M2 activation Sevoflurane suppresses microglial M2 polarization (*Pei, Wang & Li, 2017*). However, the underlying mechanism of sevoflurane-regulated microglial polarization in the aging brain remains undefined.

The transcription factor CCAAT/enhancer binding protein (C/EBP) is a family of leucine zipper transcription factors, and as the first discovered C/EBP family member, C/EBPα is mapped to 19q 13.1 on human chromosome. As a transcription factor with important biological functions, C/EBPα is distributed in several tissues and participate in the regulation of biological processes such as cell differentiation, proliferation by DNA

methylation, and energy metabolism. C/EBPα was highly expressed in the renal tissue of diabetic nephropathy rats; Sinomenine reduced IL-18 and IL-1β production by decreasing C/EBPα to improve renal glomerular endothelial dysfunction (*Zhang & Wang, 2022*). In experimental autoimmune encephalomyelitis, C/EBPα was a downstream gene of miR-124 and increased in activated microglia, contributing to CNS inflammation (*Ponomarev et al., 2011*). Notably, the expression of C/EBPα was substantially increased in M1-polarized microglia, and inhibition of C/EBPα contributed to the M2 polarization of microglia thus ameliorating intracerebral hemorrhage induced-inflammatory injury (*Yu et al., 2017*). However, whether C/EBPα play a role by regulating microglial activation in sevoflurane induced postoperative cognitive impairment has not been reported.

In this study, a POCD model of aged rats was established by anesthesia with inhaled sevoflurane to explore the effects of sevoflurane-induced anesthesia on C/EBPα protein expression levels, microglia activation levels and inflammatory factors levels in the hippocampus of aged rats, and to find potential intervention target molecules for POCD.

## MATERIALS & METHODS

### Animals and model

A total of 80 male SD rats (550–650 g, age 19–22 months) were provided by the Animal Experiment Center of Xi'an Jiaotong University (License Number: SCXK (Shaanxi) 2020-001). They were housed in ventilated cages (485*350*200 mm; temperature: 21–23 °C, humidity: 50% ± 5%) containing three to seven rats per cage with sufficient supply of food and water, at normal circadian rhythm, and adaptively housed for one week before the experiment. The rats were fasted for 12 h before the experiment and had free access to water. Using random number table method, rats were randomly divided into four groups ($n = 20$): control group, 3% sevoflurane group (Sevo), 3% sevoflurane + sh-NC group (Sevo + sh-NC), and 3% sevoflurane + sh-C/EBPα group (Sevo + sh-C/EBPα). The sequences of sh-C/EBPα and sh-NC was synthesized by Genepharma, Co., Ltd (Shanghai, China). After anesthetizing rats with 1% pentobarbital sodium solution (50 mg/Kg), 5 μL of lentivirus encoded sh-C/EBPα (5′-GTG CGC AAG AGC CGA GAT AAA-3′) at a titer of $2.0 \times 10^9$ TU/ml was injected into the left lateral ventricle of the rats and its non-targeting sequence sh-NC (5′- TTC TCC GAA CGT GTC ACG T -3′). Control and Sevo groups were injected with PBS solution 5 μL. One day after the injection, the control group inhaled 30% oxygen/air for 6 h, and the Sevo, Sevo + sh-NC, and Sevo + sh-C/EBPα groups inhaled a mixture of 3% sevoflurane and 30% oxygen/air for 6 h (*Wang et al., 2021*). Rats were placed in an anesthetizing chamber with two interfaces: one was connected to a sevoflurane vaporizer and the other was connected to a multi-gas monitor. The concentration of sevoflurane and oxygen was monitored using a gas analyzer (Dash 4000; GE Healthcare, Milwaukee, WI) during sevoflurane anesthesia. After the end of sevoflurane anesthesia, it was confirmed by blood gas analysis (Kent Scientific Corp., Torrington, CT, USA) that the arterial blood gases or pH values were not changed during the inhalation of sevoflurane anesthesia in rats. After cognitive testing, rats were sacrificed by intraperitoneal injection of pentobarbital sodium solution (85 mg/kg). This experiment was approved by the Ethics Committee of Shaanxi Provincial People's Hospital.

## Staining methods
### Hematoxylin-eosin (HE) staining

Paraffin sections of hippocampal tissues (5 μm) were obtained, deparaffinized by xylene and hydrated by adding graded concentrations of ethanol. Then, the sections were stained with hematoxylin solution for 5 min, and subsequently incubated with eosin solution for 1 min. After graded ethanol and dimethylbenzene treatment neutral resin was used to block the sections. The pathological structure of rat hippocampal tissue was observed under a light microscope (magnification, ×200; Nikon Corporation, Tokyo, Japan).

### Immunofluorescence staining

After xylene deparaffinization, the hippocampus tissue sections were washed by 0.1M PBS and microwave repaired for 15 min, blocked with 10% normal goat serum for 20 min, followed by the addition of 300 μL of primary antibodies (rabbit anti-rat Iba-1; 1:200; ab153696, Abcam, Cambridge, UK) for overnight incubation. After washing by PBS, sections were incubated with a fluorescent secondary antibody (Alexa Fluor® 488-labeled Goat anti rabbit; 1:300; ab150077, Abcam, Cambridge, UK) for 2 h in the dark, washed three times with PBS, and mounted using anti-fluorescence quencher containing DAPI. Ultimately, a confocal laser-scanning microscope was utilized to observe and photograph the sections (magnification, ×200; Olympus FV1000, Olympus, Tokyo, Japan).

## Cognitive testing
### Morris water maze task

Ten rats from each group were randomly selected for the water maze experiment 24 h after sevoflurane inhalation (*Vorhees & Williams, 2006*). The location navigation experiment was conducted from day 1 to 5, rats were placed into a pool 150 cm deep 60 cm from either quadrant facing the wall of the pool, and the platform was placed in the middle of the pool, and the time that rats found the platform was scored as the escape latency. If the rats still did not find the platform within 90 s, the rats were guided to the platform and remained on the platform for 15 s. The escape latency was recorded as 90 s, which was averaged from four measurements per day. On day 6, a spatial exploration experiment was performed, the platform was withdrawn, and rats were placed into the water from either quadrant facing the wall of the pool, and the number and swimming speed at which they crossed the platform in 90 s were recorded.

### Fear-conditioning test

The conditioned fear test can be used to examine associative learning and memory functions in rats (*Zhang et al., 2022*). Conditioned fear experiments were performed on days 12 and 13 after sevoflurane inhalation. Rats were individually placed into the conditioned fear box and allowed rats to freely explore the box for 100 s. Rats were then given three groups of stimulation. Each group was given a sound stimulus 5,000 Hz, 85 dB, 30 s, immediately followed by a shock of 0.8 Ma, 2 s. Each set of stimuli was separated by 1 min. After the stimulation ended, the rats were allowed to continue to stay in the box for 30 s. The rats were removed. After 24 h of the testing, the rats were placed back in the same box and removed 6 min later, and the freezing time of the rats during the 6 min session was

recorded. After 2 h, another box was taken with the odors in the box and the lights all differed from the first. Rats were placed into a second box and given three sets of sound stimuli: 5,000 Hz, 85 dB, 30 s, with 1 min interval between each set of stimuli. The rats' freezing time was recorded.

## ELISA

After 50 mg of frozen hippocampal tissue was added to 1.5 ml of saline and ground to a slurry using a homogenizer, the supernatant was collected by centrifugation at 12,000 rpm/min for 15 min at 4 °C, and IL-6 (SP12279), IL-10 (SP12294), TGF-β (SP12253), and TNF-α (SP12250) in hippocampal tissue was detected according to the instructions of ELISA kits (Wuhan Saipei Biotechnology Co., Ltd., Wuhan, China).

## Western blotting

After 50 mg of frozen hippocampal tissue was lysed in protein lysis buffer (containing 1% protease inhibitors) on ice to extract total protein, protein concentration was measured using bicinchoninic acid (Beyotime, Shanghai, China). Denatured protein samples were obtained at 50 $\mu$g after row separation by electrophoresis, go to a polyvinylidene fluoride membrane. After blocking treatment, 1:1 000 dilution of primary antibodies were added, respectively, and placed for overnight incubation at 4 °C. The following day, after washing the membrane, a 1:5 000 dilution of secondary antibody was added and incubated for 2 h at room temperature. After exposure was developed by chemiluminescence, the gray levels of the bands of interest were determined using image J software, and the ratio obtained from the gray levels of the internal reference GAPDH was defined as the relative amount of the protein of interest.

## RT-qPCR

After 50 mg of frozen hippocampal tissue was harvested and RNA was extracted by the Trizol method (15596026, Invitrogen, Carlsbad, CA, USA), it was reverse transcribed to synthesize cDNA (*Yang, Liu & Chu, 2020*). The mixed reaction system was preheated at 95 °C for 6 min followed by 40 cycle stages: 60 °C for 30 s after 95 °C for 10 s. The mRNA expressions of CD86 and CD206 in hippocampal tissues were detected by $2^{-\triangle\triangle CT}$ assay with U6 as internal references.

## Chromatin immunoprecipitation (ChIP) assay

ChIP assays were performed using the EZ chip kit (Millipore, Burlington, MA, USA) according to the manufacturer's protocol. DNA protein crosslinks were generated by adding 1% formaldehyde to cells and incubating for 10 min. Cross linked chromatin DNA was then sonicated to yield chromatin fragments of 200-300 base pairs that were incubated with anti-C/EBPα (Abcam, Cambridge, UK) or IgG at 4 °C. The promoter region of HDAC1 was examined by RT-qPCR.

## Luciferase reporter assay

Lipofectamine 2000 (Invitrogen, Carlsbad, CA, USA) was employed to co-transfect HEK-293T cells with the luciferase report vectors, containing promoter serial truncations of HDAC1, and vector or C/EBPα overexpression vector (pEX3-C/EBPα). After HEK-293T

cells transfection of 24 h, the luciferase activity was detected based on the manufacturer's instructions with the Luciferase Assay Reporter System (Promega, Madison, WI, USA).

## Statistical analysis

SPSS 23 statistics was used for data analysis. After the Shapiro Wilk test, normally distributed measurement data were expressed as mean $\pm$ standard deviation, and comparisons among multiple groups were performed by one-way ANOVA, while comparisons between two groups were performed by $t$-test. $P < 0.05$ was considered statistically significant.

## RESULTS

### Sevoflurane-induced anesthesia promoted C/EBPα expression in aged rats

A sevoflurane-induced anesthesia model was constructed in aged rats. HE staining showed that the morphology of neuronal cells in the hippocampus of control rats was normal, and the cells were arranged in an orderly and regular manner. Compared with the control group, the neuronal cells in the hippocampus tissue of the Sevo group were unevenly distributed and irregularly arranged, and there were a large number of pyknosis phenomena (Fig. 1A). Expression of C/EBPα protein after sevoflurane anesthesia in aged rats was examined. Our results showed that sevoflurane anesthesia-induced a significant increase in C/EBP $\alpha$ protein expression in the hippocampus of aged rats. (Fig. 1B).

### Knockdown of C/EBPα inhibited microglial activation and phenotypic changes in aged rats after sevoflurane anesthesia

Microglial activation in the aging brain causes neuronal damage in neurodegenerative diseases, promoting age-related cognitive impairment. To investigate the role of C/EBPα in microglia, short-hairpin RNA against C/EBPα (sh-C/EBPα) or corresponding negative control (sh-NC) were injected into the lateral ventricles of rats before sevoflurane treatment. The protein levels of C/EBP $\alpha$ in the hippocampus were examined in aged rats after sevoflurane anesthesia by Western blotting. The results showed that C/EBPα interference could memorably reduce the expression level of C/EBPα in hippocampal tissues of sevoflurane-induced rats (Fig. 2A). Hippocampal neurons in both the Sevo and Sevo + sh-NC groups were reduced in number and disordered in arrangement, cells shrunk and appeared conical or triangular in shape with disrupted nuclear positioning or lysis and deepened staining. In the Sevo + sh-C/EBP $\alpha$ group, the number of hippocampal neurons was increased and more regularly arranged, and most of the cells had a normal morphology with regular nuclei and individual cells shrunk (Fig. 2B). Further, we uncovered that anaesthetic surgery induced a significant increase in the number of Iba1 positive cells, that is, increased microglial activation, in the hippocampus of aged rats. After injection of sh-C/EBPα, the number of Iba1 positive cells in Sevo + sh-C/EBPα group was dramatically decreased (Fig. 2C). The function of activated microglia depends on its phenotype. We used RT-qPCR to identify two distinct phenotypic markers of microglia in the hippocampus of aged rats. Sevoflurane anesthesia significantly increased CD86 (microglial M1 type marker) and decreased CD206 (microglial M2 type marker) expression. Injection of sh-C/EBPα
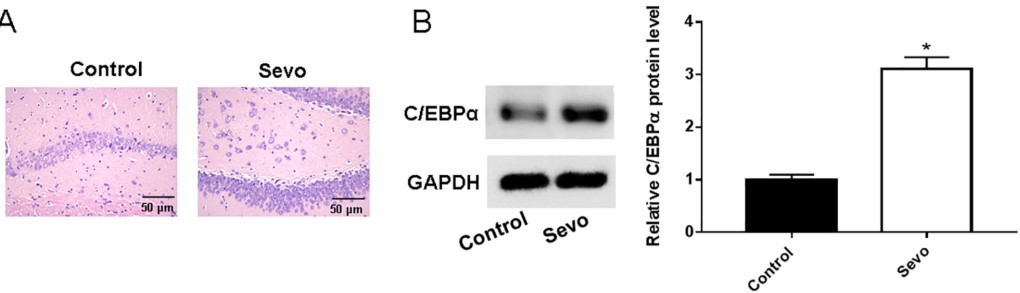

**Figure 1  Sevoflurane exacerbated the aging-induced upregulation of C/EBPα expression in the rat hippocampus.** (A) Hippocampal histopathological findings of rats in each group; (B) Expression of C/EBPα protein level in the hippocampus of rats in each group. An asterisk (*) indicates that $P < 0.05$.

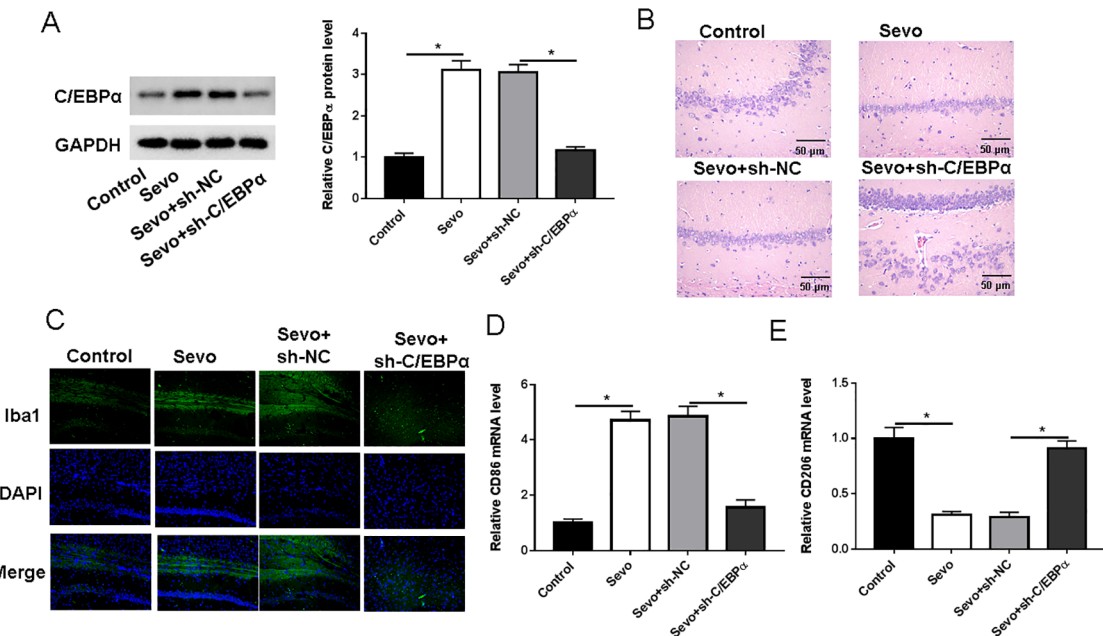

**Figure 2  Loss of C/EBPα reduced sevoflurane-induced microglial activation and polarization in aged rats.** (A) Western blotting measurement results of C/EBPα protein level in hippocampus. (B) Hippocampal histopathological findings of rats in each group; (C) Immunofluorescence measurement results of Iba1 in hippocampus. (D and E) RT-qPCR measurement results of CD86 and CD206 mRNA levels in hippocampus. An asterisk (*) indicates that $P < 0.05$.

resulted in a significant downregulation of CD86 and upregulation of CD206 in the hippocampus of aged rats (Fig. 2D and 2E).

## C/EBPα transcriptionally activated HDAC1

There is already evidence that inhibition of HDAC1 can ameliorate neuroinflammation and cognitive dysfunction in neurodegenerative models. We found the binding motif of C/EBPα by Jaspar database, which predicted its three binding sites (P1, P2 and P3) to the promoter of HDAC1 (Fig. 3A). Luciferin reporter assays showed no significant difference

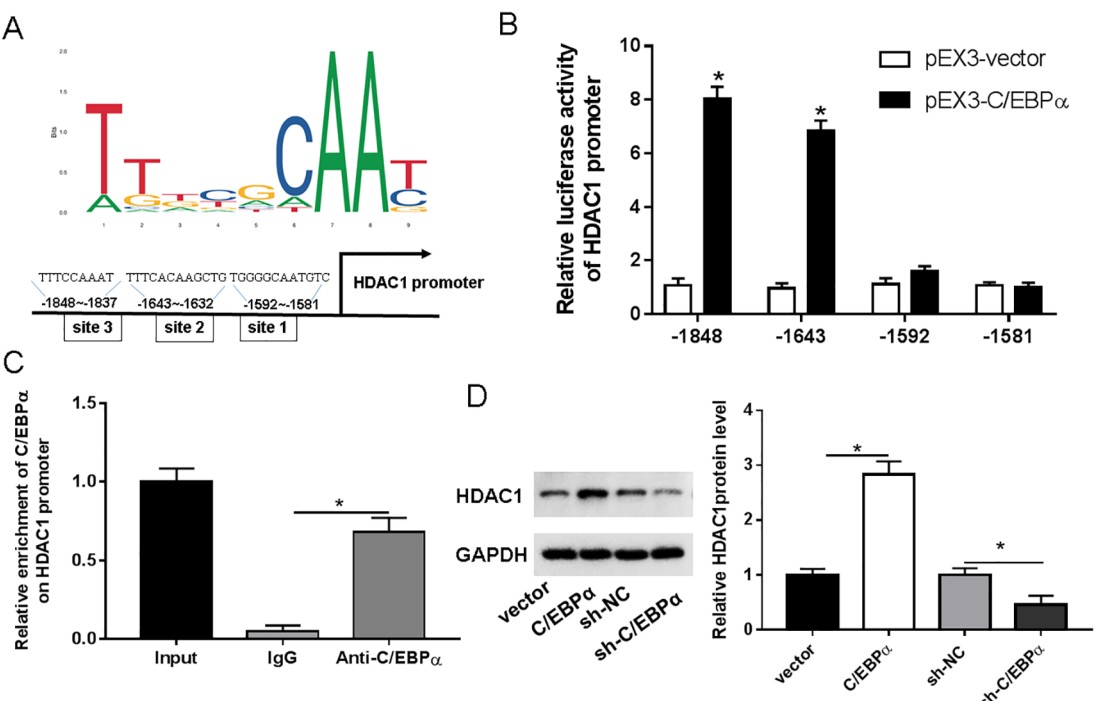

**Figure 3  C/EBPα was involved in regulating HDAC1 expression.** (A) Schematic representation of C/EBPα binding motif and C/EBPα binding sites on the promoter region of HDAC1; (B) Luciferase reporter assay measurement results of relative luciferase activities of HDAC1 promoter; (C) ChIP measurement results of the enrichment of C/EBPα on HDAC1 promoter; (D) Western blotting measurement results of HDAC1protein level. An asterisk (*) indicates that $P < 0.05$.

in the luciferase activity of the vector containing the P1 and P3 binding sites compared with the vector group, whereas the luciferase activity of the vector containing the P2 binding site was significantly higher (Fig. 3B). ChIP experiments further indicated that C/EBPα bound to the promoter region of HDAC1 (Fig. 3C). Analysis of the western blotting demonstrated that an elevation of C/EBP α also led to an efficient rise in HDAC1 expression, while interference of C/EBPα deceased HDAC1 expression (Fig. 3D).

## Knockdown of C/EBPα regulated the expression of inflammatory cytokines *via* HDAC1/STAT3 pathway in aged rats after sevoflurane anesthesia

The release of inflammatory factors by activated microglia can cause synaptic and neuronal damage in the hippocampal tissue and thus participate in POCD. To further confirm the regulation of inflammatory responses by C/EBPα in aged rats after anesthesia, we examined the production of inflammatory cytokines (IL-6 and TNF-α) and anti-inflammatory cytokines (IL-10 and TGF- $\beta$) in the hippocampus region, respectively. The expression levels of IL-6 and TNF-α in the hippocampal tissues of aged rats were increased after sevoflurane anesthesia, whereas the expression levels of IL-10 and TGF- $\beta$ were decreased after sevoflurane anesthesia. Silencing of C/EBPα inhibited the up-regulation of IL-6 and TNF-α expression and the down-regulation of IL-10 and TGF- $\beta$ expression induced

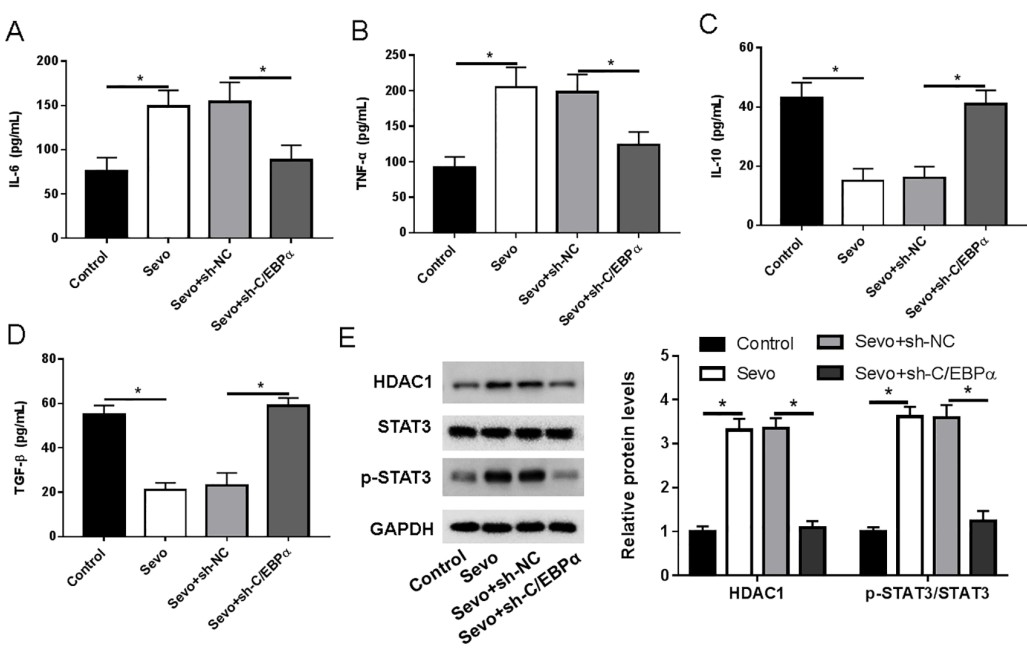

**Figure 4 Loss of C/EBPα reduced sevoflurane-induced inflammatory cytokines *via* HDAC1/STAT3 pathway in aged rats.** (A–D) ELISA measurement results of IL-6, THF-α, IL-10 and THF-β levels in hippocampus. (E) Western blotting measurement results of HDAC1 and STAT3 phosphorylation protein levels in hippocampus. An asterisk (*) indicates that $P < 0.05$.

by sevoflurane anesthesia (Figs. 4A–4D). Furthermore, activation of the HDAC1/STAT3 pathway contributes to the activation of microglia and promotes neuroinflammatory responses. The western blotting results revealed that sevoflurane anesthesia-induced a significant increase in the HDAC1 protein and STAT3 phosphoprotein levels in the hippocampus of aged rats. After injection of sh-C/EBPα, the level of HDAC1 protein and STAT3 phosphoprotein in Sevo + sh-C/EBPα group was dramatically decreased (Fig. 4E).

## Knockdown of C/EBPα prevented the impairment of cognitive function induced by sevoflurane anesthesia in aged rats

To investigate associative learning and memory abilities in aged rats, we performed a conditioned fear test in rats after sevoflurane anesthesia. The results of the study showed that rats in the sevoflurane anesthesia group had reduced freezing time in contextual fear compared with the control group, whereas no significant difference was found in freezing time in voice fear. Compared with the Sevo + sh-NC group, aged rats in the Sevo + sh-C/EBPα group showed prolonged freezing time in contextual fear, with no obvious difference in freezing time in acoustic fear (Figs. 5A and 5B). Next, we used MWM (Morris water maze) to detect the spatial learning and memory ability of sevoflurane-induced aged rats. The results of the MWM study showed that in the testing, as well as in the control group, the aged rats in the Sevo group had significantly longer escape latencies and fewer times to traverse the platform on the 4th day. Compared with rats in the Sevo + sh-NC group, rats in the Sevo + sh-C/EBPα group had significantly shorter escape

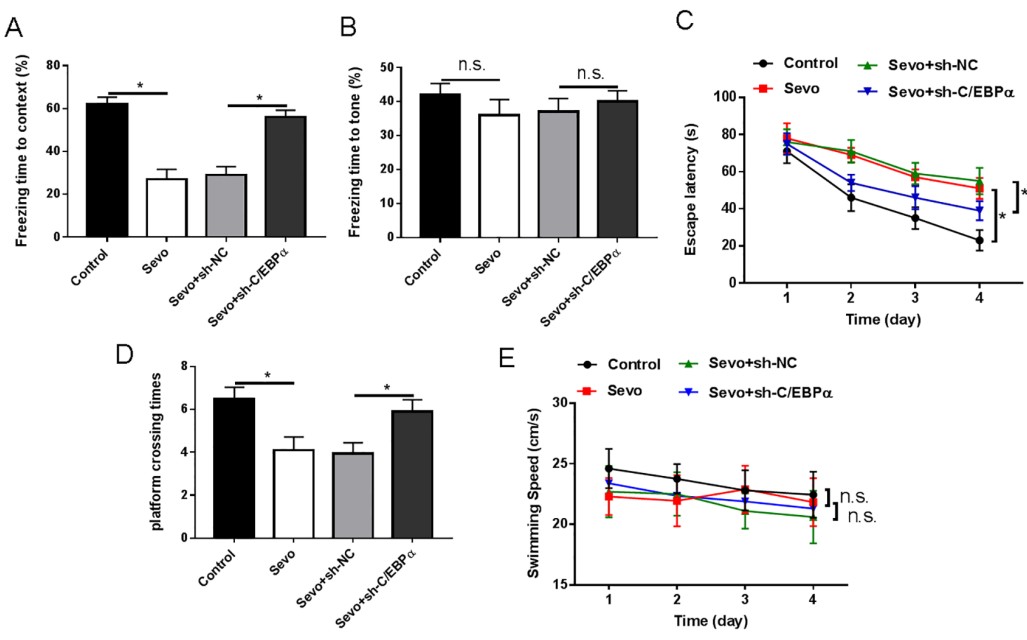

**Figure 5  Loss of C/EBPα alleviated sevoflurane-induced cognitive impairment in aged rats.** (A) FCT assay measurement results of freezing time in contextual fear of rats; (B) FCT assay measurement results of freezing time in voice fear of rats; (C) MWM assay measurement results of escape latencies of rats; (D) MWM assay measurement results of the number of platform crossings of rats; (E) MWM assay measurement results of swimming speed of rats. An asterisk (*) indicates that $P < 0.05$.

latencies and increased number of platform crossings on day 4. In addition, the swimming speed differences during the experimental period among the four groups of rats were not statistically significant (Figs. 5C–5E).

## DISCUSSION

Advantages of sevoflurane such as low blood gas partition coefficient and less respiratory irritation to patients are widely used in clinic. Sevoflurane has induced neuroinflammation by stimulating inflammatory factors such as IL-1β and TNF-α, and has also induced neur oxidative damage by activating calcium dependent proteases to produce a large amount of reactive oxygen species, which has promoted neuronal apoptosis, leading to cognitive dysfunction. Sevoflurane is widely used in clinical anesthesia and can further aggravate nervous system injury on the basis of surgical factors. Found in cases undergoing cholecystectomy that inhalational anesthesia with sevoflurane results in postoperative cognitive dysfunction in patients was more likely than intravenous anesthesia with propofol (*Geng, Wu & Zhang, 2017*). Compared to propofol anesthesia regimens, patients who underwent cardiac surgery with external circulation under sevoflurane anesthesia had a significantly higher incidence of postoperative cognitive dysfunction and risk of adverse effects (*Tang et al., 2019*). Therefore, an in-depth understanding of the mechanism of sevoflurane-induced cognitive dysfunction is of great importance for clinical better prevention and treatment of the occurrence of POCD.

Most of the current studies on the effects of sevoflurane on POCD focus on middle-aged and elderly patients. Previous studies have shown that aged rats showed impairment in cognitive function when they inhaled sevoflurane at a concentration of higher than or equal to 3% for 6 h (*Shao et al., 2020*; *Yin et al., 2022*; *Xu & Qian, 2020*). Therefore, 3% sevoflurane inhalation for 6 h was used for treatment in this experiment. After inhalation of sevoflurane, aged rats showed cognitive impairment, uneven and irregular distribution of neuronal cells in the hippocampus, increased expression of C/EBPα, activation of microglia, and elevated secretion of inflammatory factors.

Studies have shown that C/EBPs play an essential role during brain development. C/EBPα was lowly expressed in the hippocampal tissue of diabetic rats and might be one of the important molecular targets that contribute to neurodegeneration in diabetes (*Kazkayasi et al., 2013*). *Pan et al. (2013)* revealed that IL-13 significantly enhanced C/EBPα, which further promoted activated microglia death, thereby preventing cerebral inflammation and neuronal injury. In Alzheimer's disease, NF-κB was one of the substrates of C/EBP $\beta$, and the increased C/EBPβ promoted NF-κB p65 nuclear translocation, contributing to NF-κB mediated inflammatory reaction and neuronal degeneration, thus aggravating cognitive impairment of APP/PS1 rats (*Yi-Bin et al., 2022*). Our data showed that after interfering with the expression of C/EBPα in rat hippocampal tissues *in vivo*, the rats showed significantly shorter escape latency, significantly increased number of crossing the platform, significantly lower concentration of inflammatory factors in hippocampal tissues, and improved cell pathological morphology in the water maze test, suggesting that C/EBPα may play some role in sevoflurane-induced cognitive dysfunction in rats.

Histone deacetylase 1 (HDAC1) belongs to the histone deacetylase family and has the ability to regulate chromatin structure and transcription, and most importantly, it can participate in the occurrence of inflammatory responses. For example, enhancing the activity of HDAC1 contributed to the conversion of microglia M1 to M2 phenotype by decreasing KLF4 deacetylation, thus promoting neuroinflammation (*Ji et al., 2019*). *Wang et al. (2017)* reported that H19 enhanced TNF-α, IL-1 $\beta$ and reduced IL-10 by driving HDAC1-dependent M1 microglial polarization in ischemic stroke. There is also evidence that HDAC1 affected cognitive impairment by inducing neuroinflammation. Inhibition of HDAC1 improved microglia overgrowth and pro-inflammatory factor hypersecretion by upregulation of Tet2 to treat memory impairment and cognitive symptoms in aged Alzheimer's disease-related rats (*Li et al., 2020*). In anesthesia/surgery induced POCD, HDAC1 was enriched in the hippocampus, accompanied by a severe inflammatory response and learning and memory impairment (*Yang et al., 2020*). In traumatic spinal cord injury, parthenolide facilitated shift from M1 to M2 polarization of microglia by inactivation of STAT3 *via* downregulation of HDAC1 (*Gaojian et al., 2020*). *Yang et al. (2020)* uncovered that loss of HDAC1 exerted protective effects in POCD *via* partially inactivation of JAK2/STAT3 signal pathway. In current study, after inhalation of 3% sevoflurane for 6 h in aged rats, the expression levels of HDAC1 and phosphorylated proteins of STAT3 in hippocampal tissues were increased, the polarity of microglia shifted from M2 to proinflammatory M1 phenotype, and the secretion of proinflammatory factors was increased, while the secretion of anti-inflammatory factors was decreased. However,

knockdown of C/EBPα reduced the expression of inflammatory phenotype of microglia, improved neuroinflammatory response and reversed cognitive dysfunction by selectively enhancing microglial function after M2 polarization.

These findings lay a new theoretical basis for generating insights into the roles of C/EBPα and are meaningful for further clinical usage in POCD treatment. In addition, these results also suggest that sevoflurane can induce cognitive impairment in elderly patients, and alternatives can be used instead of sevoflurane when used clinically However, this experimental study did not conduct the exploration of the effects of surgical trauma combined with sevoflurane on superimposed cognitive function in rats, and *in vitro* cell experiments were not performed to jointly verify the conclusions, which has some limitations.

## CONCLUSIONS

Sevoflurane activates microglial M1 polarization and inflammatory responses, leading to hippocampal tissue damage and the occurrence of POCD by promoting C/EBPα expression. Inhibition of C/EBPα can downregulate the HDAC1/STAT3 pathway, promote microglia to M2 type transition, inhibit inflammatory responses, attenuate hippocampal histopathological and morphological damage, and improve cognitive function in aged rats. In addition, alleviating the prevention and treatment of central neuroinflammatory responses may provide new ideas for the prevention or treatment of POCD in the elderly.

### Funding
This work was supported by the Shaanxi Provincial People's Hospital Science and Technology Development Incubation Fund (2022YJY-34) and the Shaanxi Natural Science Basic Research Program (2023-JC-QN-0951). The funders had no role in study design, data collection and analysis, decision to publish, or preparation of the manuscript.

### Grant Disclosures
The following grant information was disclosed by the authors:
Shaanxi Provincial People's Hospital Science and Technology Development Incubation Fund: 2022YJY-34.
Shaanxi Natural Science Basic Research Program: 2023-JC-QN-0951.

### Competing Interests
The authors declare there are no competing interests.

### Author Contributions
- Zhao Xu conceived and designed the experiments, performed the experiments, prepared figures and/or tables, and approved the final draft.
- Xi Yao conceived and designed the experiments, performed the experiments, analyzed the data, authored or reviewed drafts of the article, and approved the final draft.

- Yikang Zhao analyzed the data, prepared figures and/or tables, and approved the final draft.
- Bo Yao performed the experiments, prepared figures and/or tables, authored or reviewed drafts of the article, and approved the final draft.

## Animal Ethics

The following information was supplied relating to ethical approvals (i.e., approving body and any reference numbers):

The Shaanxi Provincial People's Hospital approved the study.

## Data Availability

The raw data is available in the Supplemental File.

## Supplemental Information

Supplemental information for this article can be found online at http://dx.doi.org/10.7717/peerj.15466#supplemental-information.

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
