# Peer review of "C/EBPα involvement in microglial polarization via HDAC1/STAT3 pathway aggravated sevoflurane-induced cognitive impairment in aged rats"

_PeerJ, doi:10.7717/peerj.15466_

## Round 0.1 · original submission · Major Revisions

The reviewers have suggested several good comments. Please address them carefully.

·

Basic reporting

English language, and grammar, are acceptable but the writing style should be checked and improved. Some sentences are way too long and confusing as I noted below. The English language is not my native language, but in my personal opinion, the manuscript checking by a more experienced colleague would be beneficiary. The introduction and background are acceptable.

Experimental design

The article meets the scope of the journal. The research questions are meaningful and would be even stronger after the acceptance of the reviewer's suggestions. Methods need more explanation due to the lack of information about animals as I noted, for example: did the authors perform any surgeries because in some parts of the text they mentioned surgery but there was no explanation in the text above? If they did, the analgesic approach must be explained considering the ethical standards. I require explanation regarding to the preanesthetic preparation as well from the same reason.

Validity of the findings

The findings of the study are validated. Their findings are meaningful for further clinical usage. I suggested mentioning the same in the conclusion as well as the authors are anesthesiologists (clinicians).

Reviewer 2 ·

Basic reporting

The manuscript by Xu et al investigates the role of C/EBPα in POCD. The study demonstrated that depending on the presence of C/EBPα, hippocampal histopathological injury, microglial polarization, the production of inflammatory cytokines and cognitive dysfunction chances in sevoflurane-induced elderly rats. The authors further showed that the effect of C/EBPα on sevoflurane anesthesia- induced cognitive impairment is mediated by a yet unknown mechanism due the insulin receptor.

Experimental design

It is an interesting study, but there are some issues:
1. This study shows that C/EBPα knockdown ameliorates sevoflurane-induced cognitive impairment in aged rats. Therefore, it is suggested in the title to mention C/EBPα aggravated sevoflurane-induced cognitive impairment in aged rats.
2. In the introduction, it is suggested to mention the prevalence of postoperative cognitive dysfunction in patients.
3. ՙThe control group inhaled 30% oxygen for 6 h, and the Sevo, Sevo + sh-NC, and Sevo + sh-C/EBPα groups inhaled a mixture of 3% sevoflurane and 30% oxygen for 6 h. Please provide mention references for the method of establishing POCD model.
4. In line 280 and 281: Anaesthetic surgery induced a significant increase in the protein level of HDAC1/STAT3 in the hippocampus of aged rats. But, in figure 4 showed that the protein level of HDAC1 and STAT3 phosphorylation in the hippocampus of aged rats. Please changed.
5. Please provide more information about the C/EBPα and negative control: are these scramble sequences? Please provide the sequences. Moreover, how were the doses of C/EBPα/negative control selected?

Validity of the findings

6. Please provide references about morris water maze task and fear-conditioning test assays.
7. In figure 1B, and 2A, please clarify whether the statistical plots are for the C/EBPα protein or for the mRNAs.
8. The polarization of microglia and the relationship of the HDAC1/STAT3 pathway were not illustrated, so the conclusion "" improvement of sevoflurane-induced cognitive dysfunction in aged rats via the HDAC1 / STAT3 pathway "", which is logically lacking, please further specify.
9. In general, the codes of the kits used should be reported in the materials & methods section;

Additional comments

no commet

·

Basic reporting

The manuscript proposal meets professional standards in terms of language and structure. The introduction and background show very well the context of the study. The literature is well-referenced and is relevant to support the research.

However, I leave some comments for the authors in the sections different from my revision, in the hope that these suggestions will help their submitted proposal achieve publication quality.

L80: please remove an extra square bracket, before the word sevoflurane.

Experimental design

Original primary research within scope of the journal.
In general, in this section, it is required to add the percentage or concentration, working times, brand, model, version, laboratory, and country of origin, of all the drugs, reagents, solutions, antibodies, equipment, kits, and software used, as appropriate. In addition, the methods used require the support of at least one reference for each one.

L110: please add the inclusion and exclusion criteria considered for this research.

L110-111: please add the type of cage used with its respective measurements, indicate if there was temperature, relative humidity, and controlled ventilation in the rodent housing, also clarify the type of food the animals received, that is, crude protein content and metabolizable energy from the diet. When referring to water consumption, write ad libitum.

L114: Please clarify if 3% sevoflurane was the vaporized fraction or the minimum alveolar concentration (MAC) measured from an anesthetic gas monitor.

L126: Add the approval registration number.

L133: indicate the magnification at which the hippocampal tissue was observed. Also indicate characteristics of the optical microscope.

L145: This method requires a reference.

L148: please add a reference.

L160: This method requires a reference.

L181: please add a reference.

L187-189: add the amount of goat serum, antibodies, and PBS solution used. Also, clarify whether the goat serum was stored frozen, fresh, or refrigerated before use.

L193: what characteristics does the microscope have?

L210: indicate the normality analysis used. Before that, add how the calculation for the sample size was done.

Validity of the findings

The study has an impact, in addition to being novel in aspects not evaluated in anesthesia, such as the case of the evaluation of cognitive impairment induced by sevoflurane in a geriatric model.

Other comments are:
L370: Before addressing the conclusions, please add a section that includes a discussion of the limitations of this research and the perspectives for the future with the findings described.

About the figures:
In Figure 1, item A, add a brief explanation of what was observed in the histopathology of the hippocampus. Particularly the graphic included in item B is missing to indicate the measurement units.

Figure 2 presents results that are relevant, however, these lose importance because the figure is crowded. I suggest to the authors that this figure be divided into 3-4 to express the findings obtained with more order and depth.

Figures 3 and 4 (graph item E only) present the same comprehension difficulties. I suggest that the units of measurement be indicated in the graphs and that the explanation of each item presented is improved.

Additional comments

The strengths of the manuscript are:
It is written in a professional and unambiguous language.
It is a serious study supported by an ordered investigation.
It is a relevant topic for anesthesiology, particularly in the area of geriatric patients.

---

## Round 0.2 · accepted · Accept

After carefully the revision and response letter, in my opinion, this revised manuscript could be accepted.